# Ethics, Responsibility, and Sustainability in MBAs. Understanding the Motivations for the Incorporation of ERS in Less Traditional Markets

**Gaston Fornes** [1,2]📵, **Abel Monfort** [3]📵, **Camelia Ilie** [4], **Chun Kwong (Tony) Koo** [5,*] **and Guillermo Cardoza** [4]

[1] School of Sociology, Politics and International Studies, University of Bristol, Bristol BS8 1TH, UK; g.fornes@bristol.ac.uk
[2] EAE Business School, 08015 Barcelona, Spain
[3] ESIC Business & Marketing School, 08021 Barcelona, Spain; abel.monfort@esic.edu
[4] INCAE Business School, 960-4050 La Garita, Costa Rica; camelia.ilie@incae.edu (C.I.); guillermo.cardoza@incae.edu (G.C.)
[5] International Innovation Hub, Zhejiang University of Technology, Hangzhou 310014, China
[*] Correspondence: tony@zjut.edu.cn

**Abstract:** This study of Master of Business Administration (MBA) programs in regions with different history, background, legacies, and trajectories than those in the Global North aims at having an alternative view of how Ethics, Responsibility, and Sustainability (ERS) are incorporated in management education. To this end, the research uses case studies, analyzes in-depth interviews, and adopts an inductive stakeholder theory approach to identify and understand the motivations for the incorporation of the broad area of ERS in management education in relation to the schools' main stakeholders, mainly students and their employers. The analysis of the data shows that individual motivations (individual level) and an articulated and embedded mission that incorporates different stakeholders (organizational/curriculum level) are strong predictors. Local regulations and legislation, along with the requirements from international accreditation agencies (institutions/environment level) are also predictors, although not that strong to go beyond the incorporation of a Corporate Social Responsibility (CSR)-related course in the curriculum of programs. Nevertheless, these CSR-related courses (organizational/curriculum level) are powerful mediators that create, as a minimum, awareness of ERS in MBA graduates who as a consequence modify their employment objectives. The data also show that the process leading to international accreditations (institutions/environment level), the expectation by employers that MBA graduates should have an ERS mindset/skills toolkit (institutions/environment level), and a hands-on, practice-based teaching methodology (organizational/curriculum level) can act as moderators. These findings show that business schools can become ERS predictors themselves, and to achieve this they need to have a better understanding of the different roles played by the different variables. This publication is based upon work from COST Action CA18215 – China in Europe Research Network, supported by COST (European Cooperation in Science and Technology), www.cost.eu.

**Keywords:** MBA; management education; ethics; sustainability; responsibility; CSR

---

## 1. Introduction

The appearance of the book "Shut Down the Business School: What's Wrong with Management Education" by Parker [1] in 2018, and more importantly the number of reviews of this book being published in leading journals (see for example [2–4]), seem to be indicating the starting point of a new call to rethink the role of Business Schools (BS), similar to that seen after Bennis and O'Toole's

article published in Harvard Business Review in 2005 [5]. The roots that built up these calls are a little different. At the beginning of the 21st century the main trigger was the Enron case, while in the 2010s it was the 2008 financial crisis and its aftermath. However, the consequences are similar, since both situations question the practice and ethics of BS graduates and therefore the value/values of their management education [6]. In fact, in one of the reviews Davies and Starkey [2] go a little further to assert that "we appreciate creative destruction and the inevitable rise of new business models in the b-school sector".

Among the main concerns in assessing this/these value/values is what Parker [1] defines as the hidden curriculum, the idea "that the purpose of the business school curriculum is to conceal capitalism as common sense" [3] along with a "market managerialism" [4] that perpetuates "a particular form of organizing that relies on hierarchy, inequalities of status, and reward" [1] leaving "human factors and all matters relating to judgment, ethics, and morality" [5] in second place. This framework implicitly has an instrumental approach [7] which brings the "perception of a necessary tradeoff between managerial courses of action that are morally sound and those that are economically profitable" [8] which seems to be what the next generation of leaders attending BS are receiving with regard to the complex challenges faced by business and society [9]. Although it is doubtful that management education can deal with all ethical dilemmas, or that it can be made responsible for how people act outside the classroom (as their inner values are developed over their entire lives), at least it should become a place for exposure, interaction, and experiences to make cognitive and effective changes in students that have a positive influence in their approach to decision-making processes [1].

According to these works, this approach to management education seems to be more prevalent mainly in what is called the Global North, i.e., in the more traditional BS located in the US, Canada, Northern Europe and developed areas of Asia. What about newer and/or less traditional BS located in other places? Are they following the same approach? This is relevant, as out of the estimated 15,000 BS in the world, more than 50% are located in other regions outside the Global North [10]. In addition, around 15 BS every year are receiving the Association of MBA's (AMBA) accreditation for their flagship MBA programs and most of them are not located in the Global North (at the time of writing, the total number of AMBA-accredited BS, is 270+ [11]).

This bias towards the Global North in the understanding of management education has already been identified, explicitly or implicitly, in previous works (see for example [12–15]) with very few having studied BS beyond the more traditional locations (see for example [16,17]). This leaves an important gap in terms of geographical coverage, and also (and more importantly) in knowing and understanding the approach to management education of institutions with different backgrounds, legacies, and trajectories than those in the Global North. This study aims to reduce this gap. To this end, the research uses case studies, analyzes in-depth interviews, and adopts an inductive stakeholder theory approach [18] to identify and understand the motivations for the incorporation of the broad area of ethics, responsibility, and sustainability (ERS) in management education in relation to the schools' main stakeholders, mainly students and their employers. The research focuses on the following questions:

- How do predictors for ERS in management education work in BS operating in regions with different history, background, legacies, and trajectories than those in the Global North? What is the influence (if any) of the government/regulations and other institutions on building the BS identity and learning activities through ERS criteria?
- How do mechanisms (mainly moderators and/or mediators) act for the engagement of different stakeholders in management education beyond the traditional BS based in the Global North? What is the influence (if any) on individual programs at the curricular level, in particular MBAs? How is ERS included in the learning objectives and methodologies?
- How do individual motivations underpin ERS in these higher education institutions? Are they market-driven, a reflection of benevolent school values, or a personal commitment to tackle broader social challenges?

The premises are that, as a consequence of the influences from the more traditional and well-known BS in the Global North described above, on average BS located in other places have adopted a

hidden curriculum of capitalism as common sense in their management education, the relations with stakeholders and the stakeholders' salience are low, the BS are putting ERS activities/teaching in second place after market managerialism, and leaders' vision, organizational identity and pride, and equity sensitivity are also low.

In this context, this paper intends to contribute to the management education literature in the following ways: (i) by analyzing BS with different history, background, legacies, and trajectories to those in the Global North, neglected in the literature, (ii) by studying their ERS strategies/options in their MBA flagship program (including the program curriculum), and (iii) by providing a unique setting to analyze existing theories in different contexts.

The paper is structured following the inductive approach of qualitative research. The next sections describe the research context and the scope of the qualitative methodology. This is followed by a section presenting the observations and identified patterns that led to theoretical insights. This in turn is followed by a discussion section which includes contributions, limitations, and areas for future research. The article concludes with a summary and conclusions.

## 2. Research Context

The formal educational system is an appropriate arena to promote ERS because it can shape the views and attitudes of students towards sustainability and contribute to a profound social change [19]. The United Nations recognized this by developing the Principles for Responsible Management Education (PRME) [20], in which they state that "academic institutions help shape the attitudes and behavior of business leaders through business education, research, management development programs, training, and other pervasive, but less tangible, activities, such as the spread and advocacy of new values and ideas. Through these means, academic institutions have the potential to generate a wave of positive change, thereby helping to ensure a world where both enterprises and societies can flourish" [21]. There are almost 800 signatory institutions to PRME principles (at the time of this article) which represents less than 5% of the estimated BS in the world; this may be one of the reasons why criticism has been raised regarding the reality and depth of ERS incorporation in BS learning activities.

On this, Denning [22] said that, "there has been little change in the core curricula of business schools [where it is possible to see a] disconnect between what is taught and the vast ongoing societal drama under way", "despite some individual thought-leaders in BS" making strong claims to address current and emerging issues. Spender [23] added that "the bulk of ethics teaching in business schools is narrowly Aristotelian and makes no attempt to define contemporary business's special challenges". In this context, for example, McDonald [24] analyzed Harvard Business School's approach and found that they do teach ethics, although its MBA experience is biased towards a narrow analytic thinking which treats the money of companies as the most significant product of managers. In addition, Stiglitz [25] said that the resulting business managers offer a "land of PowerPoint presentations and cuddly good intentions" with feel-good clichés (like "win–win" and "make a difference") around "doing well while doing good" instead of getting deeply into what about contemporary capitalism needs to change. It is not surprising to see, then, that BS students cheat more than others [23].

The situation contrasts with the calls, first by Springett [26] and then by Kelley and Nahser [27], that education should not be neutral regarding values and ethics to which Boyce [28] added that students and teachers should go beyond their technical knowledge to question their part in starting, reproducing, and sustaining the problems and inequalities in the economy and in the society. This is because higher education institutions are important pillars of a more sustainable development in knowledge generation to inform public policy and stakeholders' engagement, in their daily operations to reduce their negative impact and maximize their positive social impact, and in the knowledge and skills transferred to their graduates who then become professionals and consumers; i.e., they can set an example during one of the most important stages in the transformation of future business leaders [12]. In other words, strengthening ethics education can help repair the relationship between business and society [29] and, within this, higher education institutions can play a proactive role in the positive transformation of societies [30].

The impression that ERS is not deeply embedded in BS learning activities can be seen as a missed opportunity considering the power and position BS have acquired in recent years; after all, BS are among the success stories in higher education over the last century [22] and probably the most significant addition to most universities in the Global North [1]. To get a better view of the current situation, Setó-Pamies and Papaoikonomou [12] reviewed existing studies about teaching ERS in BS around the world. In their comprehensive list, there are only 30 works covering both undergraduate and postgraduate levels, around 65% focused on the Global North and the remaining 35% with a worldwide focus (including around 17% in specific locations beyond the Global North); only four of them analyzed MBAs (worldwide, USA, USA and Europe, and Pakistan). Gioia [31] added that many BS and their curricula are bound by major BS rankings (most based in the Global North) and the ERS percentage in these publications is relatively small (or even non-existent).

The challenge, then, is that the learning activities in BS have a strong influence on students' values, ethics learning, and behavior [32] as this experience has a lasting influence on graduates' work and personal life [33]. However, the previous analysis seems to indicate that most BS in the Global North follow an instrumental approach in their management education, with market managerialism as the main framework guiding their curricula, low relations with stakeholders and also low stakeholders' salience. In addition, "the absence of an external stakeholder engagement process, the lack of inclusion of material impacts in reports, and the lack of institutionalization of sustainability reporting in the higher education system" are factors hindering the change process [34]. This has implications at three levels, as identified by Setó-Pamies and Papaoikonomou [12]: organizational (mainly the definition of the institution's identity-mission, vision, values in relation to sustainability criteria, as well as the establishment of a culture and the incorporation of sustainable criteria into the institution's overall strategy), curricular (mainly the incorporation of ERS in BS' programs), and instrumental (mainly the intended learning outcomes and teaching methodologies). Getting a better understanding of all these elements is the main objective of the present study.

## 3. Research Method and Qualitative Data

The research focuses on a theoretical sampling for selecting useful cases from a specified population that have the potential to replicate or extend theory, an approach that constrains extraneous variations and sharpens external validity [35,36]. Case studies were deemed appropriate to get a thorough understanding of BS and MBAs beyond the Global North along with their engagement in ERS-related education activities, relations with stakeholders, and strategies/options along with their ERS underlying mechanisms, which are the main objectives of this research. This methodology is also better suited to examining subjective features of the study (such as stakeholders' salience, intentions to engage in ERS-related learning activities, leaders' vision, organizational mission and identity, motivations, etc.) [36,37]. Additionally, case studies produce and strengthen theoretical models since reliability and validity can be kept under control [38,39].

In this context, the study is based on a deep analysis of three MBAs (the case studies) [40–42] based in Asia, Latin America, and Southern Europe (within the setting of BS offering MBAs with different history, background, legacies, and trajectories than those in the Global North). These three BS were chosen because they received the AMBA accreditation between 2016 and 2018, which means that they joined the elite 2% of top MBAs in the world [11] only in recent years and therefore, in principle, their development and growth has been less influenced by what has been happening in the Global North. In addition, within the less than 25 MBA programs that fulfill this requirement, the chosen three are the only ones appearing in at least one of the most relevant MBA rankings (i.e., Financial Times, The Economist, Bloomberg Business Week) at the time of this research. The case studies were developed by the Deans/Managing Directors of the three BS. Participant observation was chosen to get a holistic understanding of what was studied as objectively and accurately as possible, and also to get strong insights into the context where the unit of analysis operates, increasing thus the validity of the study [43].

Additionally, and mainly to minimize the potential insidership bias of participant observation [44], the research analyzes data collected through in-depth interviews with the directors of another three MBAs based in the same place as the case studies with the aim of becoming comparators and benchmarks during the analysis. These three programs were also selected following the principles of theoretical sampling [35]; they operate in the same place, they are also well-known in their local markets (therefore they are somehow competitors of the three case studies), and they were not accredited by AMBA (at the time of the research). The latter was considered relevant to avoid a potential influence from the widely followed standards in the Global North. In addition, in-depth interviews were carried out with three MBA graduates and three employers of MBA from the places where the MBAs operate (Asia, Latin America, Southern Europe). The 12 interviewees operate within similar idiosyncratic characteristics (managerial, organizational, and environmental) making the responses operative and, as a consequence, a similar contextual view was obtained [45]. A summary of the sample and its characteristics can be seen in Table 1.

**Table 1.** Sample characteristics.

| Case Studies. | # Master of Business Administration (MBA) Students | Location | Developed by |
|---|---|---|---|
| Case study 1—BS1 | 200+ | Asia | Managing Director |
| Case study 2—BS2 | 200+ | Latin America | Dean |
| Case study 3—BS3 | 250+ | Southern Europe | Dean and MBA Director |
| **Interviewees** | **Type** | **Location** | **Position** |
| CO1 | Competitor MBA | Asia | MBA Director |
| CO2 | | Latin America | MBA Director |
| CO3 | | Southern Europe | MBA Director |
| GR1 | MBA Graduate (interviewed within 6 months of finishing the program) | Asia | Senior Consultant |
| GR2 | | Latin America | Senior Manager |
| GR3 | | Southern Europe | Senior Manager |
| EM1 | MBA Employer | Asia | CEO-Multinational |
| EM2 | | Latin America | Owner-Medium-sized family business |
| EM3 | | Southern Europe | Director-Multinational |

The data collection was developed in three main stages. First, the three Deans/Managing Directors discussed, analyzed, and constructed the initial diagnosis of motivations, moderators/mediators, stakeholders' engagement, predictors, institutions, role of the government, etc. This diagnosis revealed how different players perceive the different topics impacting ERS-related activities and generated a broad framework to continue with the other interviews. The second stage involved interviews with the directors of competing MBAs, and the third stage with MBA graduates and MBA employers. This field study proved appropriate to understand the multi-dimensionality of the BS' ERS approach, the relationships between BS, directors, faculty members, students/graduates, employers, etc. as well as the predictors and outcomes. The interviews were transcribed verbatim from tape recordings, yielding 170+ pages of text. All participants were promised anonymity, the research took place between 2018 and 2019.

Similar to previous studies using similar methods (such as Kotabe et al. [46] or Valax [47]), the analysis began with a few initial assumptions and continued with comparisons between responses and coding of data which resulted in the identification of patterns, similarities, and differences. Finally, the findings were analyzed through triangulation between the case studies (internal and external), previous works, and secondary data. These processes aimed at providing internal validity. This iteration, similar to Brown and Eisenhardt [48], helped to develop the constructs and theoretical insights. The summary of the analysis of the interviews is presented in Table 2.

**Table 2.** Summary analysis of the collected data.

| Level | Question | Source | Most Relevant Quotes |
|---|---|---|---|
| Institutions/ Environment | Motivations | CO2, CO3 | The content and focus of the MBA are what the market demands, and for this we benchmark what others do in the national market . . . considering market what candidates seek CO2.<br>Candidates look first for general management, then finance, and then marketing when looking for an MBA CO2.<br>Students have to understand the companies in their program; from this point, I think what school is doing is that they're looking on the market and we have directors, we are looking on the market. And what is the market demanding from the students now CO3.<br>To fulfil the requirements from the regulator, we have some competencies that are required to be covered CO3. |
| | Underlying mechanisms (mainly mediators/moderators) | CO1, CO3 | The new Dean used to work in XXX University, so we benchmark our programs with its BS CO1.<br>It is a pity from regulator's requirements that we don't have much flexibility to change the program, this is coming from the country's system. The pity is that you are not so flexible when you want to change the core mandatory program nor the mandatory courses that they have to take CO3. |
| | Predictors/ Institutions/ Government | CO1, CO2, CO3, EM2 | The details of the program did not happen from the government side, but rather from the market side. As a business school, we need to know what kind of future talent is required in the market CO1.<br>We have always thought that AMBA provides us with a very good framework, on how to develop our program in a very efficient way, in a sustainable way CO1.<br>Large companies are increasingly focused on CSR, but smaller companies are more interested in survival and make money CO2.<br>In my experience of more than 20 years, I have never seen CSR in the requirements for new employers CO2.<br>CSR is less important than the other points that they look. I mean, the mindset is still not there. Companies look for some specific competences but CSR is not part of them CO3.<br>The ERS part in the recruitment stage is difficult to measure, but we rely heavily on references, after all this is a small world EM2. |
| Organizational/ Curriculum | Motivations | CO1, CO2, CO3, EM1, EM2, EM3 | The new Dean really brings some very new ideas, very advanced cutting-edge ideas and see that we need to shift from the traditional one to the most advanced one . . . he really has a very good view for the future CO1.<br>There is no requirement from the University on the content and focus, it is mainly on the market and the experts; the program director is responsible to take the lead on this CO2.<br>The mission of the school is preparing professionals suitable for the market so they can do better in their careers CO3.<br>We have some values for the success of the company EM1.<br>In every interview we show the six values of the company and see how they can be compatible with the candidate EM2<br>During the interview we try to see how candidates can fit the values of the company. The personal fit with us is worth around 30% in the recruitment process EM3. |

**Table 2.** *Cont.*

| Level | Question | Source | Most Relevant Quotes |
|---|---|---|---|
| | Underlying mechanisms (mainly mediators/moderators) | CO1, CO3, GR1, GR2, GR3, EM1, EM2, EM3 | We can only build up this kind of sense in the CSR course, so that they know what would be the general requirement of business in the future CO1. <br> There are many definition and ways of thinking. There's no one. In the case of CSR or sustainability, is there any, do you have a definition of this, because this is very broad CO1. <br> CSR is included in the program in a specific compulsory course (one of ten) and also implicit in some of the other compulsory courses. <br> In strategic management, even in marketing, things like that, you have to be ethical, you have to take a corporate social responsibility mindset, but also in the structure. And you know the things we cover in the courses, sometimes we don't have time to repeat those things in different courses. So we focus more on the people and soft skill management part, than on specific things like CSR or ethics, ethical behavior, and that in each of the courses CO3. <br> As the program director I don't request the professors to specifically add the CSR in strategic management CO3. <br> It's difficult to measure how much the school can contribute to the person's ethical or sustainable views CO3. <br> The school can help to shape a bit, but in the one or two years that they are at the school, it wouldn't change drastically how the person is CO3. <br> The first thing I looked for was what kind of company resources and network. And then second, the design of the curriculum GR1. <br> In the CSR course I learned that discrimination is part of corporate social responsibility, and the importance of diversity and respect, the differences among cultures. I also got the importance of anti-corruption GR1. <br> In the interview for my current job at [a major internationally renowned consulting firm] they did not say anything about sustainability. But when I received the welcome letter it was in the front GR1. <br> I checked many programs and the contents are basically the same GR2. <br> From the Nordic faculty that taught the CSR courses I could learn how this is done in the leading countries, and compare with the situation in my own country GR2. <br> To improve the MBA learning I would propose to see first-hand how companies are working on CSR GR2. <br> I was checking MBAs with more focus on strategic management, finance/accounting, or consultancy GR3. <br> CSR is important, but not essential to me in and MBA GR3. <br> I think what is more important for an MBA is to have the perspective of the corporate responsibility but from the company side, rather than only a course GR3. <br> We expect people to be accountable . . . we check this during the interviews EM1. <br> From candidates we expect they understand how we do business. The first is we need to make money, this is the first point to be sustainable; then the second point is to be responsible in hiring, purchasing, etc., and also in the community EM1. |

**Table 2.** *Cont.*

| Level | Question | Source | Most Relevant Quotes |
|---|---|---|---|
| | Underlying mechanisms (mainly mediators/moderators) | CO1, CO3, GR1, GR2, GR3, EM1, EM2, EM3 | In my experience ... most students in XXX MBA where I taught do not have even knowledge of sustainability, or it was too simple EM1. Students need to make a linkage between business, between markets, between customers, and finally there is the integration on sustainability and sustainable market EM1. When they are students it is too early for them to understand, it is real life, and then they are facing management challenges ... if they can lead in this direction they can give lots of benefits to society EM1. The way to teach students, the way to inspire students make the difference EM1. If MBA students do not have the experience, it is a waste of money ... if they cannot apply what they are learning there is no real value EM1. Sustainability for us is to be long term in the market EM2 I would like to see a more social and environmental elements embedded in the development of persons during their education EM2 We have specific universities/BS in the list of preferred suppliers due to their education in ERS EM3. |
| | Predictors/ Institutions/ Government | CO3, GR1, GR2, EM1, EM2, EM3 | Companies expect people to bring good values, but they are not measuring their CSR credentials in the hiring. This is basically what we are seeing CO3. What I see is that CSR is less relevant than other factors when they choose MBA CO3. First is the faculty and their cultural background, the diversity of cultural background of the faculty. And secondly, their professional experience" GR1. After the MBA, when I read about a company I want to work, I can analyze these points [related to ERS]. Whether they have a CSR program, and whether if ethics is important for them or not. So if the ethics is not important for them, I prefer to say no to that company, because it also depends on me GR2. The most important thing we look for in candidates are analytical thinking, diversity awareness/cultural differences, results focus/ability to deliver EM1 My company received the Sustainability Business Award from the China-Europe Chamber of Commerce EM1. There is no big difference between one university and another university EM1. We look for a proactive person, who can also be entrepreneurial ... and analytical persons with data, budgets, dashboards ... EM2. The MBA has been commoditized, and not all universities prepare their graduates in analytical skills EM2. In graduates from specific universities, like XXX, YYY, ZZZ, we focus on the technical element, and we tend to recruit from the ones that are renowned in creating good professionals EM2. The most important things when recruiting are personal skills; within this the professional/technical skills are the most important points EM3. We are leading in sustainability in the industry, for example changed the entire fleet to electric vehicles for 3000 daily routes, and this puts us in what people want EM3. |

**Table 2.** *Cont.*

| Level | Question | Source | Most Relevant Quotes |
|---|---|---|---|
| Individual | Motivations | CO1, CO2, CO3, GR1, GR2, GR3 | All our goal when we develop a new program is to fulfill the requirements of the market CO1. For the current student/manager, CSR is in second place, the first place is taken by technical skills . . . . in our courses most of students are executives with several years that want to certify their experience. CO2. We still need to strengthen the culture in our country to have a wider focus on CSR, especially in education where graduates are agents of change CO2. From my experience what they see outside the company is not important, they do not give much importance to CSR when they hire someone CO3. [BS1] was the only MBA institute that had an entire global faculty GR1. Before the CSR course [in the MBA] I thought it was only a field of study, but then I realized it is actually a joint effort by different stakeholders and how we stand, what kind of code of behavior for sustainability, and what kind of social responsibility should we take into consideration GR1. The MBA gave me the awareness . . . now I want to work for a company with a clear focus on social responsibility, even if I sacrifice income GR1. I looked for an MBA outside my country to have an international experience, and then for expertise in marketing GR2. For me, to have that kind of [CSR] classes were really, really good, because they give me a really different point of view GR2. Now I can decide to work for a company with responsibility credentials, even if it pays less. I need to get some money [to pay the 50% of my MBA], but not to get money I will do anything GR2. I looked for an MBA with international exposure, beyond the academic part GR3. If I take a job in a company with lower sustainability credentials but with a higher salary, then the thing is that I do not know if it will still be operating in five years. I need money now to repay the MBA. GR3. |
| | Underlying mechanisms (mainly mediators/moderators) | CO1, CO2, GR2 | When we say that this is a general requirement that should be incorporated in every course, then professors decide how their courses include this kind of contents CO1. The definition and approach for CSR teaching is decided by each professor CO2. I wanted a really good MBA, a good MBA with a reputation. For me, it is important that the degree could be used all over the world GR2. What really clicked with me was more the picture of my country against the teachers' country, that totally changed my mind GR2. |
| | Predictors/Institutions/ Government | CO1, CO2, CO3, GR3, EM2 | Companies usually say, we do not want to hire MBAs because they just have the theories, but they do not know how to solve the problem when a new challenge comes CO1. The fees of the MBA program are also a predictor of the level of the students/graduates, usually those that can afford an MBA already have a good position; this means that we teach those that do not need it CO2. I think it's a cycle that companies now are getting more aware of. They are giving more importance. So they will incorporate more and more this in the future I think for the new hires that they will do CO3. Though CSR is important, I don't think companies are taking it as a priority. They are not taking that as their priority even though they say that. Although personally I think that it's important GR3. A person that fails with truth or shows low quality is fired EM2. I see ERS as an investment, when you help people there is a reduction in loses, the more we help the community the better the result EM2. |

This combination of elements, selection of case studies (including comparators and benchmarks), in-depth interviews with internal and external players, coding of data, triangulation and pattern matching, aims to provide reliability, internal construct, and external validity to support the findings of the analysis [49]. This research can be conceived as a so-called critical case as it carries a strategic importance in relation to the general problem being addressed [38,50,51]; therefore, it can contribute to set the basis for further studies of the preliminary results which have been obtained [52].

## 4. Case Analyses

The analyses of the three case studies follow the structure of the research questions; they aim at identifying and understanding the motivations for the incorporation of ERS in the BS learning activities. First, the focus is on the institutional/environmental elements that can potentially be predictors. Second, the assessment is on the curricula, learning objectives, and how ERS is incorporated into the MBA programs. Finally, the attention goes to individual motivations inside organizations for including ERS in the learning activities.

### 4.1. Case Study 1 (BS1)

The University holding the program was founded in the 1920s and is run by the local government; it is the largest in the city (and one of the largest in the country) with around 3000 faculty members and 37,000 students in undergraduate, graduate, and doctoral programs. The first MBA program was launched in 2003 and since then it has been run by an MBA center (BS1 in Table 1), which has some autonomy for the design and implementation of its strategic and resourcing plans, although the academic strategy is set by the University. The University's core education goals are framed by the all-round and whole-person development vision and mission set by its founding president which permeate all learning activities in the institution; in fact, it is also the first and core element of the MBA program. The University, "through the continuous improvement of its distinctive all-round and whole-person education model, aims at providing society with graduates characterized by having a global vision, civic consciousness, compassion, innovative mindset, and ability to execute to meet future challenges" (University's website).

BS1 (and the University) are bound by the academic regulations set by the country's Ministry of Education; within this MBA programs follow the requirements of the National MBA Education Supervisory Committee which also grants licenses to universities to offer this kind of program. This regulatory body sets the duration (minimum two years), number of credits (minimum 45), and the compulsory core courses in the curriculum which should include a final project/thesis. Within this, until 2018, the Committee required nine core courses in MBA programs; none of them was related to ERS. The situation has changed from the 2019 academic year in that the Committee has added five new compulsory core courses (making a total of 14), two related to the broad area of ERS (Corporate Governance, and Business Ethics, Sustainability, and Professional Ethics). The requirements mainly concern the names of the courses and their length; this means that MBA providers/faculty members can decide the approach and methodology to deliver them. There is no limit/requirement on the additional courses/learning activities that can be included/added by universities/business schools to the core curriculum.

Within this framework, BS1 designed an MBA curriculum to have a strong positioning based on the unique education model set by the University (shown above). To do this, they followed four main criteria: (i) global/local focus, (ii) practical orientation, (iii) employability, and (iv) requirements of international accreditation agencies and rankings. The ERS element was developed within global/local focus; they analyzed trends in the global environment, in particular what was happening in Western markets (mainly the US and Europe) at the time of launching the program (2002–3), and designed a compulsory course on Compliance. The course has been evolving since then; after a few years it became Business Ethics and Corporate Governance, and then Business Sustainability and Corporate Governance. The course is currently in the fourth stage of its evolution, adding technology and

technological change/evolution to business sustainability and corporate governance. This course (along with the others offered on top of the national compulsory curriculum) has been reviewed annually by a committee formed by practitioners, academics, and other stakeholders; when the AMBA application started, this review became more formal and systematic, which was a big help from the accreditation process.

In addition, in 2009 (after only five years of launching the MBA program), BS1 was granted membership of the UN's PRME, one of the first in the country. This was a major step towards strengthening the positioning of this young program in the academic community and among employers. It was also a strong lever to incorporate an ERS perspective in the teaching across the MBA curriculum (i.e., beyond the specific ERS course presented above) as professors were asked to observe PRME's principles; this was important as there were no strong government guidelines (until 2019, ERS was not in the compulsory curriculum as shown above). However, this membership proved not to be very relevant for recruitment; candidates said they looked for good ranking positions, good employment opportunities, and good teaching more than the ethics and sustainability element in MBA education. They valued it when they were told about its existence, but in most cases this was not strong enough to change their decision.

Additionally, in the early years of the program BS1 developed a personal transcript to complement the traditional academic transcript issued by higher education institutions. This personal transcript includes a description of the student's behavior during the program beyond academic performance and measures elements like good citizenship, contributions to the class and program, etc. It was a major success, and in fact the university and other renowned universities in the city started to adopt a similar practice, and potential employers began to ask for this personal transcript, especially multinational companies with strong codes of ethics.

*4.2. Case Study 2 (BS2)*

BS2 is a not-for-profit institution founded more than 50 years ago with the first MBA awarded in 1969. Every year around 400 students participate in its full-time and part-time Master's programs and 11,000 in executive education. The School has a strong link to Latin America and its sustainable development, with a mission to develop the region's "leaders with a global perspective and to create applied knowledge . . . particularly in the areas of competitiveness and sustainability, entrepreneurship, and women's leadership". This mission permeates all the activities in the School by: (i) including courses related to ERS in its programs (degree- and non-degree seeking), (ii) adding an ERS element in the research efforts and resources, and (iii) having an ERS approach when dealing with different stakeholders, in particular with governments and companies.

The School holds the authority to award degrees as an international organization. In practice, this means that BS2 has the freedom to design and implement its curricula, methodologies, and other teaching and learning activities; for this purpose, it uses the requirements and guidelines from internationally renowned accreditation agencies as benchmarks. In this context, and following the School's mission, the ERS element is included in the programs' curriculum in both mandatory and elective courses. It is also present in two other distinctive areas of the School's activities, research, and impact events.

In both the full-time and part-time MBA programs, ERS is included in the curriculum as a result of the School's mission which requires a permanent screening of the latest trends and emerging themes in the business context. To achieve this, the School is divided into two main areas of activities, mission centers (mainly learning activities), and impact centers (mainly knowledge development and thought leadership). On the latter, the directors work on three elements—impact, applied research, and consulting—which results in a constant updating and upgrading of the programs in cross-collaboration with mission activities. For example, they have created a simulator that measures the social progress of a country and its impact on the lives and living standards of its citizens. In addition, there is a systematic benchmark with other MBA providers, locally and internationally (mostly those in the

Financial Times ranking), to meet and/or exceed the standards of the leading institutions. Additionally, after every teaching activity there is an assessment of the impact of the learning on participants and on the impact generated by the whole program. All this has resulted in an embedded culture of sustainability in the School and as a consequence in everything it does.

In addition, collaborating with global partners has increased the awareness of the ethical challenges faced by managers running complex organizations in different parts of the world, especially with issues related to corruption. Some of these challenges are different from those usually found in Latin America, and for this reason the approach has been on stressing the importance of ethics and on making participants think like good citizens (and not only like successful businesspersons). Integrity is the keyword used. As part of this, ethics and integrity have been included in communications, strategy, leadership and managerial competencies, operations, and sustainability courses. They have also been included in the analysis of case studies for most of the courses in the curriculum and have become part of the assessments of learning. Additionally, in recent years, teaching methods have encouraged live cases (to complement the traditional written cases) in which senior managers/practitioners share their challenges (including ERS-related) and exchange points of views in a more direct way with full-time and part-time degree program students. In addition, some courses require the application of what they learn in class; for this they work with local communities which creates a strong engagement with a high impact in the region.

Additionally, at the beginning of the program students produce an individual development plan working along with a coach. The process starts with their own values and their own purpose in life and compares how aligned they are with doing good and reaching their objectives without exercising power or doing harm to people, organizations, or society. From this point onwards, everything they do in the program is analyzed against this development plan; for this they have goal sheets and sessions that run in parallel for a continuous assessment. For example, every decision they make during the program receives feedback on the impact on people, teams, organizations, etc. The key question is "am I being an ethical and/or resonant leader?", the answers lead to a reflection on the students' own integrity. The whole process is based on a leadership model sought by the program (within its learning objectives) that starts with a 360-degree individual analysis and concludes by the end of the program, also with a 360-degree assessment. As a result, students can reflect on their critical thinking capabilities and also on the moral/ethical frameworks to make personal and business decisions. Interestingly, most graduates return for further coaching sessions after having spent some time working after completing their degrees.

Finally, the School organizes events, forums, and conferences on sustainability, it participates actively in a global network of researchers on sustainability, professors carry out projects for governments on sustainability, etc. These efforts have helped the School to attract candidates seeking an MBA program with a clear focus on values, sustainability, and integrity; around one-third of every intake says that this is the main reason to join the program (a percentage above the average).

### 4.3. Case Study 3 (BS3)

BS3 is a private not-for-profit institution founded more than 50 years ago. Around 6000 students take part in its academic programs every year (undergraduate and graduate) and 15,000 participate in its executive education activities. Its first MBA program was launched in 1986. The School has a strong focus on human values inspired by the beliefs of its founders (who are still running it), in fact an important element in its mission is " ... the diffusion of a business ethics culture at both national and international level ... " (School's website). Within this framework, BS3 actively promotes ethical behavior, good governance, and sustainability in the main areas of the Institution; social, business, research, and academic. In the latter, it offers courses related to business ethics, sustainability and corporate social responsibility at both UG and PG level along with extra-curricular activities and awards recognizing students' initiatives and work.

BS3 is bound by academic regulations from a 2011 national decree. This piece of legislation owned by the Ministry of Education lists the intended learning objectives of the different degree levels, foundation/associate, undergraduate, postgraduate, and doctoral. Related to ERS, for postgraduate programs it states that upon completion graduates should "know to assess and select the proper scientific theory and relevant methodology of the field of study to form an opinion from incomplete or limited information including, when needed and pertinent, a reflection on social or ethical responsibility linked to the solution proposed in each case". In practice this means that higher education institutions have a wide space to decide and design what and how they include ERS-related activities in the curriculum of PG programs, also " . . . including, when needed and pertinent . . . " seems to indicate that ERS in Master's programs (including MBAs) may not need to be an essential part of all the work carried out by students.

Traditionally, the three MBA programs in the School had a compulsory course related to ERS built around the Corporate Social Responsibility (CSR) field of study. There was no common methodology nor a common definition/focus of CSR; these were left to the professors delivering the course to decide and implement (in around 15 cohorts per year) provided that they fulfill the intended learning objectives of the program. This situation changed first with the run-up to the AMBA accreditation and then again with the program review for one of the MBAs. The first change—with the intention to improve: (i) the convergence among programs and cohorts and (ii) the demonstration of achievements/assurance of learning also among programs and cohorts—was the creation of groups led by a course leader and formed by all the professors teaching this specific course (CSR in this case); they became responsible for the development of a common syllabus, mainly the content, key definitions, and demonstration of achievements/assessments. Individual professors kept the freedom to deliver the course in the most suitable way for each of them provided that they follow the commonly agreed content, focus, and assessments. The majority of professors across all the courses welcomed the initiative and saw it as an improvement in the coordination and external validity of the courses; less than 10% opposed the initiative on academic freedom grounds (this was not the case nor the intention, each group had complete freedom to develop their syllabus within the framework set by the learning objectives of the program). As a consequence of this change, on top of strengthening the academic quality assurance in the program, BS3 stepped up to offer in its MBA programs a sharpened and deeper view of CSR that resulted from a debated and agreed perspective among professionals in the field.

The second change was part of the periodic review (five-year cycle) of one of the MBA programs. A working group was formed with different stakeholders to define what the renewed MBA would be; among other relevant outcomes the group proposed to deepen three key elements of the program's intended learning objectives: (i) increasingly globalized business environment, (ii) increasingly digitalized business operations and markets, and (iii) increasing importance of being/becoming a good citizen. These three elements were already captured in the different parts of the program (mainly in specific/traditional courses like innovation, digital business, CSR, etc.); the intention was to take the next step and make the program gravitate around them. This required a new design of each building block/course to provide a dynamic, horizontal/transversal, decision-making approach to the learning process in the MBA that superseded the one based on functions/disciplines. To achieve this, the first step was the joint definition of each of the three elements and then professors worked in groups of two or three (based on the closeness of their courses, for example accounting and finance) to incorporate them transversally across the whole curriculum. To measure their achievement, it was required that between 5 to 10 points (out of 100) of the assessment were about each of the three elements; i.e., between 15% and 30% of the achievements across the program were about them. The 5% to 10% related to ERS (being/becoming a good citizen) were about the following proposed question to be included in each assessment: "how do the proposed recommendations make the company/organization/you a better [global corporate] citizen?"; this means that every single piece of work developed by students in the program should have this question answered and marked. In addition, a new ten-hour workshop (Becoming a Global & Ethical Manager) was added to the curriculum at the very beginning of the

program to frame the whole MBA experience, leaned towards ERS. This resulted in an innovative MBA program that both candidates and employers praised for dealing with the key and more relevant challenges faced by businesswomen/men in the current and near future business environment, in particular the emphasis on ERS, and also for the strong focus on decision-making rather than on functions. The members of a peer review panel of a major international accreditation body said that this MBA is an example of international best practice.

## 5. Interviews Analysis (See Table 2 for a Summary of the Collected Data)

The approach to the interviews and subsequent analysis started with some assumptions as stated above following Eisenhardt [35] and Yin [40]. The report follows the same structure of the case studies to improve comparability and analysis, first institutional/environmental elements, second organization and curriculum, and third individual motivations.

### 5.1. Competitor MBAs

The directors of the competing MBA programs stated that fulfilling the local regulations was the first step in the design and delivery of the curriculum; in relation to ERS, for example, CO3 said that "to fulfil the requirements from the regulator, we have some competencies that are required to be covered" (see also intended learning in Case Study 3). Once these requirements are fulfilled (which in general allow for the expected academic freedom), the three directors said the next step was to research what the market is seeking and demanding in order to design the curriculum, usually in a systematic way through the programs' directors and/or the marketing area in their institutions. Asked about the market demand for ERS, the directors of CO1 and CO3 implied something similar to CO2, "candidates look first for general management, then finance, and then marketing when looking for an MBA"; i.e., ERS seems not to be within the top priorities, in fact for CO2 "the first place is taken by technical skills" and for CO3 "people and soft skills management". CO1 added that a major driver for curriculum change in the School in recent years has been the arrival of a new Dean, s/he used to work in XXX University [a top-ten institution in the world], so we benchmark our programs with its BS".

In these three schools, ERS is included as a specific and compulsory course, mainly following the requirements set by the regulations and usually framed by the CSR field of study. In this context, in general there are no guidelines on the ERS definition or scope, nor on the teaching methodology ("as the program director, I do not request the professors to specifically add CSR"—CO3). In addition, one of the respondents said that the curriculum is so broad that "we do not have time to repeat those things [ERS] in different courses" (CO3), even if the three acknowledged that there is a need to behave ethically and have a socially responsible mindset. On the other hand, the three MBA directors indicated that there does not seem to be a demand from companies for ERS-related specific skills. CO3, for example, said that "companies look for some specific competences but CSR is not part of them, . . . they do not give much importance to CSR when they hire someone". In this context, CO2 affirmed that "in my experience of more than 20 years, I have never seen CSR in the requirements for new employers" to which CO1 added "companies usually say, we do not want to hire MBAs because they just have the theories but they do not know how to solve the problem when a new challenge comes" when talking about ERS.

Having said this, the three interviewees reflected on the importance of ERS in business education, although based on what they are obtaining from the market they did not seem very optimistic on the potential positive impact that their programs can create in this area. CO1, for example, stated that "we can only build up this kind of sense in the CSR course, so that they know what would be the general requirement of business in the future" to which CO3 added "it is difficult to measure how much the school can contribute to the person's ethical or sustainable views . . . the school can help to shape a bit, but in the one or two years that they are at the school, it would not change drastically how the person is". CO2 went a little further to assert that "the fees of the MBA program are also a

predictor of the level of the students/graduates, usually those that can afford an MBA already have a good position; this means that we teach those that do not need it".

*5.2. MBA Graduates*

Graduates agreed with the program directors that their main motivation when searching for MBA options was not ERS. GR1 said that "XXX was the only MBA that had an entire global faculty; I analyzed the diversity of cultural background of the faculty. And secondly, their professional experience". GR2 said "I looked for MBAs outside my country to have an international experience, and then for expertise in marketing . . . I wanted a really good MBA, a good MBA with a reputation. For me, it is important that the degree could be used all over the world". GR3 said "I looked for an MBA with international exposure, beyond the academic part", "I was checking MBAs with more focus on strategic management, finance/accounting, or consultancy". Interestingly, GR2 reflected on the fact that s/he "checked many programs and the contents are basically the same". These motivations are somehow expected; they decided to make a big investment in education to develop their career; the mentioned areas are traditionally associated with high-flying business professionals.

The MBA programs of the three graduates have a compulsory course on CSR as one of the functions/fields of study within the curriculum, i.e., ERS was not a cross-program, transversal element. However, even if this was only something like 5% of the contact hours of the program, the three stressed the high impact that this course had on how they see business after the MBA. GR1, for example, said that "before the CSR course [in the MBA] I thought it was only a field of study, but then I realized it is actually a joint effort by different stakeholders" and also "I learned that discrimination is part of corporate social responsibility, and the importance of diversity and respecting the differences among cultures. I also got the importance of anti-corruption". GR2 said that the CSR course was an eye-opener, especially to have benchmarks on what others are doing ("for me, to have that kind of [CSR] class was really, really good, because it gave me a really different point of view"; "from the Nordic faculty that taught the CSR course I could learn how this is done in the leading countries, and compare with the situation in my own country"; "what really clicked with me was more the picture of my country against the teachers' country, that totally changed my mind"). GR3 complemented these comments by saying that "[companies] are not taking [ERS] as their priority even though they say that; although personally I think that it is very important".

Probably the most revealing (and positive) impact of this CSR course can be seen in what they looked for in potential employers after graduation. The most unexpected comment (because of the original motivation of seeking a well-paid corporate career) was that they were willing to get a lower income if this means joining a firm with a stronger ERS track record. GR2 made it very clear, "after the MBA, when I read about a company where I want to work, I can analyze these points [related to ERS]. Whether they have a CSR program, and if ethics is important for them or not. So if ethics is not important for them, I prefer to say no to that company, because it also depends on me . . . Now I can decide to work for a company with responsibility credentials, even if it pays less. I need to get some money [to pay the 50% of my MBA], but not to get money I will do anything". In a similar line, GR1 highlighted the importance of the MBA in her/change of focus when looking for a job, "the MBA gave me the awareness . . . now I want to work for a company with a clear focus on social responsibility, even if I sacrifice income". GR3 approached this by taking a longer-term perspective, s/he stated that "if I take a job in a company with lower sustainability credentials but with a higher salary, then the thing is that I do not know if it will still be operating in five years". In addition, the three agreed that to improve the take up of ERS-related skills they "would propose to see first-hand how companies are working on CSR" (GR2) and therefore "have the perspective of the corporate responsibility from the company side, rather than only a [CSR] course" (GR3).

*5.3. MBA Employers*

The interviewed employers confirmed what graduates and MBA directors suspected: ERS is not a priority when looking for candidates. For example, EM1 said that "the most important things we look for in candidates are analytical thinking, diversity awareness/cultural differences, and results focus/ability to deliver", then s/he added "from candidates we expect they understand how we do business". EM2 complemented these comments by saying that "we look for proactive persons who can also be entrepreneurial, and analytical persons with data, budgets, dashboards". Additionally, EM3 stated that "the most important things when recruiting are personal skills; within this the professional/technical knowledge are also important points". As can be seen, ERS was not in the top priorities in the first opening question, but then when asked specifically if they incorporate ERS in the expected skills/mindset of candidates, the three included it although not in the top position. EM1 said "the first is we need to make money; this is the first point to be sustainable, then the second point is to be responsible in hiring, purchasing and also in the community . . . we expect people to be accountable". EM2 replied in a similar line, "sustainability for us is to be long term in the market . . . in every interview we show the six values of the company and see how they can be compatible with the candidate". EM3 continued in this line, asserting that "during the interview we try to see how candidates can fit the values of the company. The personal fit with us is worth around 30% in the recruitment process".

This relative low relevance of ERS in the recruitment process was not expected, especially because the three companies have relatively good ERS credentials (this was known only at the moment of the interview; it was not a criterion to be included in the sample). The week before the interview, Company 1 received the Sustainability Business Award from the China-Europe Chamber of Commerce, Company 2 claimed to be one of the most socially responsible companies in its country, and EM3 said "we are leading in sustainability in the industry, for example we are changing the entire fleet to electric vehicles for 3000 daily routes". This relative low relevance can be explained by the feeling/perception during the interviews that having an ERS mindset was a given, something that is expected to be essential for today's professionals and therefore not needed to be made explicit in the definition of job profiles. In fact, during the conversations, the three interviewees put, spontaneously, this ERS mindset at the same level of honesty/integrity; for example on this EM2 said "a person that fails with truth or shows low quality is fired . . . the ERS part in the recruitment stage is difficult to measure, we rely heavily on references, after all this is a small world".

Having said this, the three expressed that more still needs to be done. EM2 stated that "I would like to see more social and environmental elements embedded in the development of persons during their education", and EM1 shared that in her/his "experience . . . most students in XXX MBA, where I taught, do not have even knowledge of sustainability". To improve this, EM1 proposed, "students need to make a linkage between business, between markets, between customers, and finally there is the integration on sustainability and sustainable markets" which is in line with the suggestions from graduates to have first-hand experience on how companies are engaging in ERS. The challenge, said EM2, is that "the MBA has been commoditized", and EM1 expressed the complementary comment that "there is no big difference between one university and another university". Because of this, added EM3, "we have specific universities/BS in the list of preferred suppliers due to their education in ERS". Finally, EM2 as the owner of the company, concluded that "I see ERS as an investment, when you help people there is a reduction in losses, the more we help the community the better the results".

## 6. Discussion and Implications

This section is developed around the themes identified in the case studies and interview analysis. To draw conclusions from the data, the analysis selected the 'best' from a list of plausible explanations of ERS options/strategies based on considerations of epistemic virtues such as interest, novelty, and plausibility with regard to MBA education in regions with different history, background, legacies, and trajectories than those in the Global North [53]. The underlying meanings arising after completing

the study allow for the identification of a set of themes as predictors, mediators, and moderators [54] for the incorporation of the broad area of ERS in management education (outcome).

### 6.1. Predictors: Regulations/Accreditations, School's Mission, and Individual Motivations

The analysis of the curricula shows that regulations/legislation/accreditation requirements/PRME membership are predictors (antecedents of ERS actions and policies) as expected and in line with previous researches (for example Godemann, Haertle, Herzig and Moon [20]). Although they do not seem to be playing a strong role other than incorporating a CSR-related course in the programs as shown by CO1, CO2, and CO3. Predictors with stronger impact seemed to be: (i) an articulated and embedded mission and vision that incorporates the BS' different stakeholders (like CS2), and (ii) the individual motivation or personal drive/push/vision/interest of the program's director/BS' authorities (as in CS1 by including ERS when it was not a requirement or by developing the personal transcript, or as in CS3 with the incorporation of ERS transversally in the program and assessments, for example); the three case studies incorporated ERS across their learning activities beyond the regulations/legislation/accreditation requirements framed by the culture and economic context of the BS [12]. The market-driven approach does not seem to be a predictor; it appears to be too broad and not accurate as seen with CO1, CO2, and CO3. BS2, on the contrary, presented a good example of a deep engagement and high impact with specific stakeholders within this broad idea of the market.

### 6.2. Mediator: ERS-Related Learning Activities

Both graduates and employers said that all MBAs look the same as the curricula of the different MBAs analyzed for this study are very similar, on average and on the surface, to those in the Global North. The requirements by international accreditations agencies seem to maintain this homogeneity, for example AMBA (the accreditation achieved by the three case studies) specifically lists 13 elements (mainly knowledge and understanding of organizations, their context, their stakeholders, and how they are managed) that need to be included, as a minimum, in the curriculum of MBA programs to be accredited (criterion 7.3. of AMBA's MBA Accreditation Criteria (https://www.mbaworld.com/-/media /files/accreditation/mba-criteria-for-accreditation.ashx?la=en)). The local legislation and/or regulations in the three countries do not do much to change this situation, whether the compulsory courses for Case Study 1/CO1 or the intended learning objectives for Case Study 3/CO3, they basically replicate the structure of MBAs seen in the Global North. This puts ERS as one of the several elements to be taught/acquired in an MBA (ERS is one of the 13 points to be covered for AMBA) usually within the CSR field of study; similar to what McDonald [24] found at Harvard Business School (one of the most respected program in the Global North) as presented above.

This is not a problem, especially seeing the positive impact of these ERS courses as reported by graduates, key stakeholders for BS. However, this raises another question, which is whether this improved/upgraded understanding of ERS is similar to that reached in other areas of the MBA (like finance or marketing) as a result of completing a certain program of study. In other words, is it only an increased awareness (as GR1 put it) of the broad area of ERS comparable to the achievements in other subject areas or a real/deep change in the professional and personal behavior/mindset of these future business leaders? More importantly, do students still see a tradeoff between morally sound and economic profitable courses of action as Weitzner and Deutsch [8] observed? These questions imply that ERS-related courses in MBAs are mediators, i.e., underlying mechanisms/processes that relate activities with outcomes.

### 6.3. Moderators: Teaching Methodology, Accreditation Processes, and Employers' Expectations

The CSR-related course, according to both graduates and employers, is not enough at least in the way it is currently provided. It needs to be strengthened by creating better and stronger links among business, markets, and ERS (as EM1 strongly suggested) and also by giving students first-hand experience and real-world exposure to ERS-related managerial challenges as the graduates stated.

Similar findings were reported by Jabbour et al. [55] and Peet et al. [56]. This would also help to change the image that ERS is not relevant, as expressed by graduates when they were looking for an MBA to continue their business education. Then, the teaching methodology is a moderator that influences the quality of outcomes. The accreditation processes are also a moderator. On top of the requirement to include an ERS-related course (by AMBA in this case, or by the local regulations), CS1 and CS3 used the accreditation process as a lever to improve/strengthen the whole functioning and learning experience of their programs (i.e., beyond ERS). As such, it influenced the quality of the outcomes. In addition, the expectation by employers that an ERS mindset is essential in today's (and future) professionals can also be considered a moderator; a deep embeddedness in the program's curriculum would also have a strong influence on the quality of the outcomes (none of the six MBAs in the sample acknowledged this expectation). This finding also contributes to the debate on whether ERS belongs to one subject (i.e., a stand-alone course) or to several ones (i.e., embedded into several courses, Rusinko [57]). A summary of the findings can be found in Figure 1.

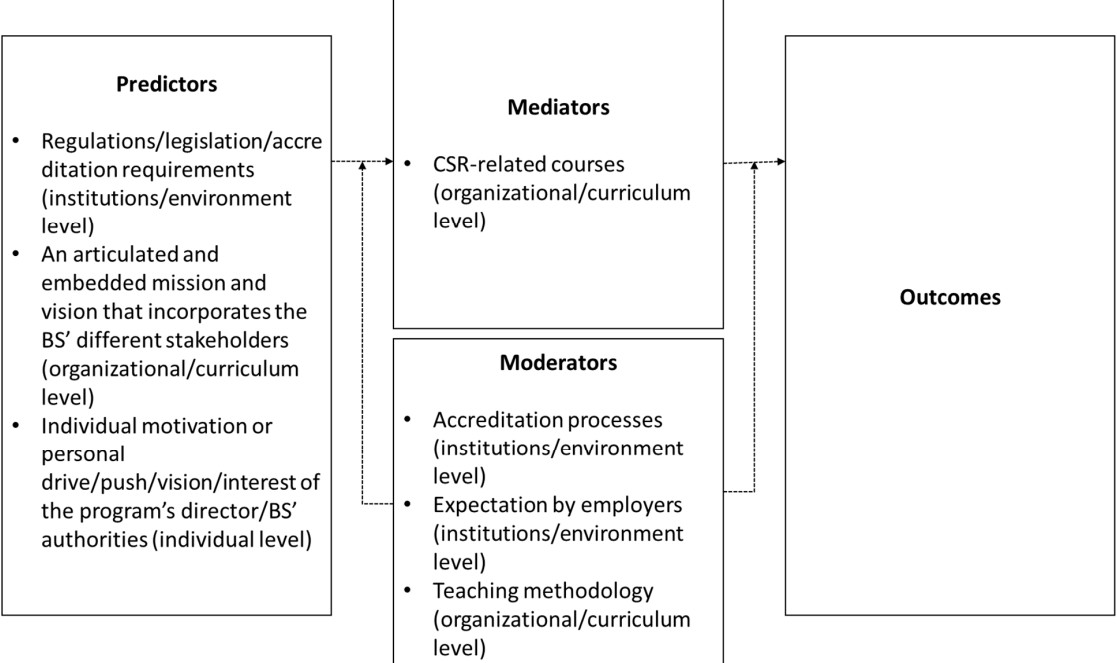

**Figure 1.** Ethics, Responsibility, and Sustainability predictors, mediators, and moderators in Master of Business Administration education.

## 7. Limitations and Future Research

Findings from three case studies are not easily generalized; due to this, several measures were taken to provide reliability, internal construct, and external validity and as a consequence the results can contribute to set the basis for further studies. In future research, the findings can be strengthened by: (i) expanding the analysis to a wider sample of BS (and probably from other regions), (ii) conducting complementary ethnographic research and focus groups, and (iii) building a standardized questionnaire focused to assess the different variables involved, and their relationships, from a quantitative standpoint [58,59]. In addition, there are areas not covered in this research that are worth investigating, for example (i) the potential predictor role of media and society to incorporate an ERS skills toolkit in management education, (ii) the climate crisis and the need of business to be more proactive in minimizing their environmental footprint, and (iii) how the expected stronger ethical and environmental standards will be included in the curriculum of MBAs.

## 8. Contributions and Conclusions

This study makes contributions to improve the understanding and practice of the internal strengths and limitations of Business Schools in relation to ERS. First, the design/adoption of a teaching methodology that offers first-hand experience and real-world exposure would better position future business leaders to deal with ERS-related managerial challenges. Second, BS authorities can focus their efforts on deepening and strengthening the ERS component in the institutions' mission, and more importantly on how it permeates the learning and transformational activities across both the curricular and extra-curricular activities in the school. Third, BS leaders can empower those within the school in the position and with the motivation to drive ERS-related learning initiatives. Fourth, schools can use the accreditation processes/PRME memberships as levers to implement changes that lead to graduates with a clearer ERS mindset/stronger ERS skills toolkit.

The findings also make contributions to theory development; the most relevant involve multilevel integration. Individual BS leaders motivations (individual level) and clear BS missions (organizational level) can supplement low stakeholder engagement and minimum ERS standards/requirements from local governments, and the expectation by employers (institutions/environment level) that MBA graduates should possess an ERS mindset/skills toolkit which can drive and strengthen ERS-related learning activities. The alignment of the three elements (leaders' motivations, BS' mission, and employers' expectations) creates a link among the three levels of activity that strengthens the positioning of BS as agents of change mediating between the main stakeholders, employers and students.

More broadly, the findings contribute to extend the understanding of how ERS is incorporated in MBA education, in particular in regions with different history, background, legacies, and trajectories than those in the Global North. Different from the original assumptions, some schools have embedded ERS as a key element in the transformation of their MBA students (whether as a result of the leaders' individual motivations or through the deployment of a clearly BS articulated mission). In these schools the relation with stakeholders and the stakeholders' salience are relatively high (mainly to embed ERS in their programs), and ERS on average is treated at the same level as other management functions (even in schools where ERS is not a transversal, cross-program element) [57,60]; local regulations/legislation and/or international accreditation requirements play a part in the latter [61,62].

In addition, the expectation by the main stakeholders, candidates/students/graduates and employers, that developing an ERS mindset and/or acquiring an ERS skills toolkit is a must-have in MBA graduates, creates an opportunity for BS to become predictors themselves and as a consequence to position as strong agents of change; a place for exposure, interaction, and experiences to make cognitive and effective ERS-related changes in students [63,64].

All in all, this study of MBA programs in regions with different history, background, legacies, and trajectories to those in the Global North with the aim of having an alternative view of how ERS is incorporated in management education, provides new insights into how to improve the outcomes for internal and external stakeholders. Of particular interest are: (i) the individual motivations (individual level) and (ii) clear missions (organizational level) that can supplement low social engagement and minimum standards/requirements from local governments to strengthen ERS-related learning activities, and (iii) the expectation by employers (institutions/environment level) that MBA graduates should possess an ERS mindset/skills toolkit. The alignment of the three elements creates a link among the three levels of activity, institution/environment, organizational/curriculum, and individual, which opens opportunities for BS to position themselves as agents of change, mediating between the key stakeholders. These findings highlight the need to continue the study of ERS-related education. BS have an opportunity to become predictors themselves and to achieve this it is necessary to have a better understanding of how key variables are playing a role.

**Author Contributions:** All authors contributed similarly to the development of the research and reports.

**Conflicts of Interest:** The authors declare no conflicts of interest.

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
