# Peer review of "Ethics, Responsibility, and Sustainability in MBAs. Understanding the Motivations for the Incorporation of ERS in Less Traditional Markets"

_sustainability, doi:10.3390/su11247060_

Round 1

Reviewer 1 Report

The paper is well-written. References are synthesized appropriately. References are extensive. Motivation for the study is clearly explained e.g., a gap in the literature. Sampling frame is clearly explained. Findings are well-covered.   

Author Response

Thank you for the positive comments, we worked hard to have a strong paper.

Reviewer 2 Report

Weaknesses of the paper: interpreting research results

The topic is interesting, but the paper has serious flaws and is accepted only after major revisions. I have some thoughts for possible improvement which the authors may consider:

Such described study could be not perceived as a “…original article…”. The paper is more analytical than scientific. Please, transparently explain the results of the research.

Author Response

Thank you for the feedback. To address the reviewer’s concerns we have done the following:

Reviewed the entire paper making improvements along the text both in content and wording. Incorporated a new column in Table 2 showing the sources of the data to increase the transparency of the data analysis. Improved the explanation of the research methods to make it clearer how the qualitative analysis was carried out, also the “limitations and future research” section has been rewritten to strengthen this point. Reviewed and rewritten the “contributions” and “summary and conclusions” sections, they are now merged to improve clarity..

On top of this and to ensure that the results are transparently explained, we double checked the data analysis against established qualitative methods publications. We found that the article follows similar patterns established by previous qualitative research when it aims to answer research questions related to "how" and "why" in unexplored or nascent areas (Eisenhardt & Graebner, 2007) such as the inclusion of ERS in MBAs. In addition, the research carries out triangulation of data by establishing specific questions to the interviewees, consulting with experts in the field, and comparing results with secondary sources following what was established by Baxter & Eyles (1997). Also, the paper is consistent with Yin (2008) on how to develop case studies.

In addition, we analyzed this research against similar investigations published in reputable journals, for example Bingham & Eisenhardt (2011), Gloria & Ding (2008) or Hervas-Oliver, Lleo, Cervello (2017). All these articles describe the analysis of interviews and then extract the main categories or dimensions, similar to our paper. In our work this was done by proposing a set of statements based on the analysis in the first part of the paper; this can be seen in Section 6. In addition, the explanation of the results is presented as clear as possible in the reviewed Section 8 where the main contributions and conclusions are explained in detail, and also in relation to previous works.

Reviewer 3 Report

The idea of this research is interesting, but I recommend the following:

I recommend that the abstract be explained in detail: MBA, CSR, ERS and BS.
In the abstract I recommend introducing the Methods sections that highlight the applied methodology.
The final finding in the abstract is not clear, I recommend the rewrite, focused on the subject.
I recommend correcting Keywords, capital letters.
Because the article is too long, I recommend that the data presented in the tables should not be the same as those presented in section 5. Interviews analysis.
I recommend that the Summary section be deleted because it repeats the ideas presented above, it is sufficient to remain only the Conclusions section.
Lines 625: The first sentence is not clear, I recommend rewriting.
I recommend that the Discussion section should have references to bibliographic sources in correlation with the results of this study, only 3 bibliographic indices are mentioned.
The number of interviewees is not clear, I recommend clarification.
Also do not mention the period of accomplishment of this research.

Author Response

Comments and Suggestions for Authors

The idea of this research is interesting, but I recommend the following:

Thank you very much for your constructive feedback.

I recommend that the abstract be explained in detail: MBA, CSR, ERS and BS.

Explanations for these acronyms have been included in the Abstract following the suggestion from the reviewer.

In the abstract I recommend introducing the Methods sections that highlight the applied methodology.

The abstract has been changed following the suggestion from the reviewer.

The final finding in the abstract is not clear, I recommend the rewrite, focused on the subject.

The final sentence in the abstract has been modified following the suggestion from the reviewer.

I recommend correcting Keywords, capital letters.

Keywords have been capitalized following the suggestion from the reviewer.

Because the article is too long, I recommend that the data presented in the tables should not be the same as those presented in section 5. Interviews analysis.

The article, at around 9,500 words (without considering tables and references), has a length similar to that found in other qualitative researches published in reputable journals. Also, we analyzed papers using similar research methods and most follow the same structure; i.e. a section with the analysis of the interviews and an appendix (a table in this case) presenting a summary of the data collected from the different sources.

Having said this, and with the aim of improving the triangulation/internal construct, we have added a new column to Table 2 indicating the sources of the data.

I recommend that the Summary section be deleted because it repeats the ideas presented above, it is sufficient to remain only the Conclusions section.

The summary section has been deleted following the suggestion from the reviewer. Also, the “contributions” section was moved to the end with a new concluding paragraph. This section is now called “contributions and conclusions”.

Lines 625: The first sentence is not clear, I recommend rewriting.

The sentence has been changed following the recommendation from the reviewer.

I recommend that the Discussion section should have references to bibliographic sources in correlation with the results of this study, only 3 bibliographic indices are mentioned.

Several references have been added along with some new text to frame them following the recommendation from the reviewer.

The number of interviewees is not clear, I recommend clarification.

The number of interviewees has been included following the recommendation from the reviewer.

Also do not mention the period of accomplishment of this research.

The period of the research was included following the recommendation from the reviewer.

Reviewer 4 Report

In their article, the Authors aim to define the role of ESR - Entrepreneurial Social Responsibility - in management education, particularly at the MBA level, to develop the relationship between entrepreneurship and Corporate Social Responsibility (CSR).

The Authors, through their analysis of the data, show that individual motivations and an articulated and embedded mission involving different stakeholders are a strong indicator of how much corporate social responsibility, as well as dynamic consideration and response to issues beyond the narrow economic and technical boundaries The company's legal requirements for achieving social and environmental benefits, as well as traditional economic benefits, raise awareness of ERS in the curricula of MBA graduates who can change their employment goals. See: lines 11-30.

Therefore, they are looking for predictors for ERS in management education in BS that operate in regions that are not part of the so-called Global North. See: line 76 forward.

They reinforce the role of individual motivations to underpin whether ERS is market-driven and possibly passively subsumed by students, or, on the contrary, by actively engaging in personal involvement of students. See: in particular lines 84-86.

In their introduction, the Authors earlier pointed out that BS (Business Schools) are widespread in the so-called global North. See: lines 57-59. In any case, the most progressive and technologically competitive part of the world, where individualism and independent relations predominate, shows that the other parts of the world must also develop the broader domains of ethics, responsibility and sustainability (ERS). See: lines 65-70.

The paper intends to contribute to management education literature in the following ways:

by analyzing BS with a different history, background, legacy, and trajectory than those in the Global North that have been neglected in the literature, by examining their ERS strategies / options in their MBA flagship program (including curriculum) and by providing a unique environment for analyzing existing theories in different contexts. See: lines 93-97.

The method proposed by the Authors for carrying out their analysis is an inductive approach with qualitative characteristics. See: lines 98-99.

In their research context, the authors attach great importance to highlighting the UN-sponsored Principles for Responsible Management Education (PRME), which is recognized worldwide only by a very small part of the esteemed BS. See: lines 106-115.

It is followed by a criticism of the fact that MBAs in the education of education are often misinterpreted only in fake pieces or in clichéd teaching methods. See: lines 116-127.

The Authors then seek a more stylish brokering of BS approaches in the so-called Global North, which usually sail the educational themes from the perspective of their own institutional rankings and a market-oriented orientation of ESR courses, rather than the perspective of curricula and curricula the importance of stakeholders.

The continued commitment of companies to behave ethically and contribute to economic development, while improving the quality of life of workers and their families, as well as the local community and society in general, and their focus on the long-term sustainability of companies analyze three MBAs as case studies based in Asia, Latin America and Sothern Europe. See: line 176 forward. The authors also conducted in-depth interviews with three MBA graduates and three MBA employers in the same locations.
They finally used triangulation to test the internal validity of interviews. See: lines 211-217.

The first case study (BS1) has underlined the authors' continued improvement of a distinctive all-round and all-person education model aimed at equipping society with a global vision, civic consciousness, compassion, innovative thinking and the following Characteristics are characterized ability to master future challenges. See: lines 236-293.

The second case study (BS2) provides an educational model that uses "integrity" as a keyword for communication, strategy, leadership and management skills, business and sustainability courses. See: lines 294-356.

The third case study (BS3) looks at BS education as the most important and relevant challenge for entrepreneurs in the current and near future of the business environment. For Features See: lines 357-425.

Predictors, mediators, results in management training, ERS latent variables are then shown in lines 432-555, such as:

the embedded mission involving various stakeholders (organization / curriculum level); the ERS Mindset / Skills Toolkit (Institutions / Environment Level); a practical, practice-oriented teaching method (organizational / curriculum level); something lies above what is regulated and expected by the community, or above the industry norm.

They combine (a) a model without covariate (no covariate), (b) a model with direct paths of covariate to each factor (covariate → latent), (c) a model with direct paths from the covariate to all factors and some Manifest variables that allow measurement without invariance or DIF (covariate → latent + manifest), and (d) a model with direct covariate paths to manifest variables but not to latent variables (covariate → manifest). See: lines 704-711.

Whether the Author would like to readdress the analysis with three samples of the same size (N=200), their final analysis model: Covariate → Latent, Covariate → Latent + Manifest, No Covariate and Covariate → Manifest could be improved by the effect size, which is calculated by subtracting the mean and dividing the result by the pooled standard deviation. The resulting effect size, dCohen, represents the difference between the groups in terms of their common standard deviation.

Kind Regards.

Author Response

Comments and Suggestions for Authors

In their article, the Authors aim to define the role of ESR - Entrepreneurial Social Responsibility - in management education, particularly at the MBA level, to develop the relationship between entrepreneurship and Corporate Social Responsibility (CSR).

There seems to be a misunderstanding in this comment, as the ERS in the paper refers to Ethics, Responsibility, and Sustainability.

The Authors, through their analysis of the data, show that individual motivations and an articulated and embedded mission involving different stakeholders are a strong indicator of how much corporate social responsibility, as well as dynamic consideration and response to issues beyond the narrow economic and technical boundaries The company's legal requirements for achieving social and environmental benefits, as well as traditional economic benefits, raise awareness of ERS in the curricula of MBA graduates who can change their employment goals. See: lines 11-30.

Therefore, they are looking for predictors for ERS in management education in BS that operate in regions that are not part of the so-called Global North. See: line 76 forward.

They reinforce the role of individual motivations to underpin whether ERS is market-driven and possibly passively subsumed by students, or, on the contrary, by actively engaging in personal involvement of students. See: in particular lines 84-86.

In their introduction, the Authors earlier pointed out that BS (Business Schools) are widespread in the so-called global North. See: lines 57-59. In any case, the most progressive and technologically competitive part of the world, where individualism and independent relations predominate, shows that the other parts of the world must also develop the broader domains of ethics, responsibility and sustainability (ERS). See: lines 65-70.

The paper intends to contribute to management education literature in the following ways:

by analyzing BS with a different history, background, legacy, and trajectory than those in the Global North that have been neglected in the literature, by examining their ERS strategies / options in their MBA flagship program (including curriculum) and by providing a unique environment for analyzing existing theories in different contexts. See: lines 93-97.

The method proposed by the Authors for carrying out their analysis is an inductive approach with qualitative characteristics. See: lines 98-99.

In their research context, the authors attach great importance to highlighting the UN-sponsored Principles for Responsible Management Education (PRME), which is recognized worldwide only by a very small part of the esteemed BS. See: lines 106-115.

It is followed by a criticism of the fact that MBAs in the education of education are often misinterpreted only in fake pieces or in clichéd teaching methods. See: lines 116-127.

The Authors then seek a more stylish brokering of BS approaches in the so-called Global North, which usually sail the educational themes from the perspective of their own institutional rankings and a market-oriented orientation of ESR courses, rather than the perspective of curricula and curricula the importance of stakeholders.

The continued commitment of companies to behave ethically and contribute to economic development, while improving the quality of life of workers and their families, as well as the local community and society in general, and their focus on the long-term sustainability of companies analyze three MBAs as case studies based in Asia, Latin America and Sothern Europe. See: line 176 forward. The authors also conducted in-depth interviews with three MBA graduates and three MBA employers in the same locations.
They finally used triangulation to test the internal validity of interviews. See: lines 211-217.

Three MBA program director from competing schools were also interviewed.

The first case study (BS1) has underlined the authors' continued improvement of a distinctive all-round and all-person education model aimed at equipping society with a global vision, civic consciousness, compassion, innovative thinking and the following Characteristics are characterized ability to master future challenges. See: lines 236-293.

The second case study (BS2) provides an educational model that uses "integrity" as a keyword for communication, strategy, leadership and management skills, business and sustainability courses. See: lines 294-356.

The third case study (BS3) looks at BS education as the most important and relevant challenge for entrepreneurs in the current and near future of the business environment. For Features See: lines 357-425.

Predictors, mediators, results in management training, ERS latent variables are then shown in lines 432-555, such as:

the embedded mission involving various stakeholders (organization / curriculum level); the ERS Mindset / Skills Toolkit (Institutions / Environment Level); a practical, practice-oriented teaching method (organizational / curriculum level); something lies above what is regulated and expected by the community, or above the industry norm.

The comments below do not seem to have relation with our paper. For this reason we do not include any reply to the reviewer here.

They combine (a) a model without covariate (no covariate), (b) a model with direct paths of covariate to each factor (covariate → latent), (c) a model with direct paths from the covariate to all factors and some Manifest variables that allow measurement without invariance or DIF (covariate → latent + manifest), and (d) a model with direct covariate paths to manifest variables but not to latent variables (covariate → manifest). See: lines 704-711.

Whether the Author would like to readdress the analysis with three samples of the same size (N=200), their final analysis model: Covariate → Latent, Covariate → Latent + Manifest, No Covariate and Covariate → Manifest could be improved by the effect size, which is calculated by subtracting the mean and dividing the result by the pooled standard deviation. The resulting effect size, dCohen, represents the difference between the groups in terms of their common standard deviation.

Round 2

Reviewer 2 Report

The paper is accepted in present form.

Reviewer 3 Report

no comments